# Cenobamate (YKP3089) and Drug-Resistant Epilepsy: A Review of the Literature

**DOI:** 10.3390/medicina59081389

**Published:** 2023-07-28

**Authors:** Jamir Pitton Rissardo, Ana Letícia Fornari Caprara

**Affiliations:** Medicine Department, Federal University of Santa Maria, Santa Maria 97105-900, Brazil; ana.leticia.fornari@gmail.com

**Keywords:** cenobamate, YKP3089, Xcopri, Ontozry, epilepsy, seizure, focal, generalized, drug resistant, antiseizure medication

## Abstract

Cenobamate (CNB), ([(R)-1-(2-chlorophenyl)-2-(2H-tetrazol-2-yl)ethyl], is a novel tetrazole alkyl carbamate derivative. In November 2019, the Food and Drug Administration approved Xcopri^®^, marketed by SK Life Science Inc., (Paramus, NJ, USA) for adult focal seizures. The European Medicines Agency approved Ontozry^®^ by Arvelle Therapeutics Netherlands B.V.(Amsterdam, The Neatherlands) in March 2021. Cenobamate is a medication that could potentially change the perspectives regarding the management and prognosis of refractory epilepsy. In this way, this study aims to review the literature on CNB’s pharmacological properties, pharmacokinetics, efficacy, and safety. CNB is a highly effective drug in managing focal onset seizures, with more than twenty percent of individuals with drug-resistant epilepsy achieving seizure freedom. This finding is remarkable in the antiseizure medication literature. The mechanism of action of CNB is still poorly understood, but it is associated with transient and persistent sodium currents and GABAergic neurotransmission. In animal studies, CNB showed sustained efficacy and potency in the 6 Hz test regardless of the stimulus intensity. CNB was revealed to be the most cost-effective drug among different third-generation antiseizure medications. Also, CNB could have neuroprotective effects. However, there are still concerns regarding its potential for abuse and suicidality risk, which future studies should clearly assess, after which protocols should be changed. The major drawback of CNB therapy is the slow and complex titration and maintenance phases preventing the wide use of this new agent in clinical practice.

## 1. Introduction

Epilepsy affects more than seventy million individuals worldwide, corresponding to an age-standardized prevalence of 621.5 per 100,000 people [1]. Approximately 3 million adults and almost 500,000 children in the United States have epilepsy [2]. Increased life expectancies and more people surviving events that can lead to epilepsy are expected to raise the number of people with epilepsy [3]. The estimated annual costs in the United States of acute seizure care are around USD 12.5 billion [4]. In this context, the burden of drug-resistant epilepsy (DRE) is believed to be significantly higher due to the number of antiseizure medications (ASMs) used concomitantly and the possible high incidence of adverse events [5,6].

Seizure freedom is a primary goal in the treatment of epilepsy. Only half of the individuals with epilepsy will become seizure-free with their first ASM [7]. Also, more than one in every three patients with epilepsy will have uncontrolled seizures despite adequate management and anticonvulsant therapy [8]. In this context, failure to achieve sustained seizure freedom with the rational use of two anti-seizure drugs administered alone or in combination defines drug-resistant epilepsy [9].

Uncontrolled epilepsy, compared to epilepsy in general, is associated with ten-to-fifteen-fold more frequent mortality secondary to traumatic injury, drowning, suicide, and sudden unexpected death from epilepsy (SUDEP) [10]. Also, some types of childhood epilepsies are related to neuronal damage leading to epileptic encephalopathy, resulting in lifelong disabilities [11]. In addition, low employment rates and lower high school graduation rates can hinder individuals with epilepsy from reaching their maximum potential [12]. Therefore, poor control of seizures can lead to a higher risk of experiencing physical and psychological disorders, causing worse healthcare outcomes, increased healthcare needs, and decreased quality of life [13].

Many new ASMs have been discovered during the last three decades, with more than twenty new ASMs approved [14]. In this context, these new anticonvulsants have improved the spectrum of side effects, increased routes of administration, and reduced the severity of epilepsy, leading to better compliance and treatment adherence [15,16]. But, there were no significant changes in the proportion of individuals affected by DRE. Interestingly, the prevalence of DRE in the 1980s was sixty-three percent, and in 2014 this number was sixty-four percent [17].

In November 2019, the Food and Drug Administration approved the commercialization of cenobamate by SK Life Science Inc., (Paramus, NJ, USA) for adult focal seizures. The European Medicines Agency approved cenobamate by Arvelle Therapeutics Netherlands B.V. (Amsterdam, The Neatherlands) in March 2021 [18]. Cenobamate is a medication that could potentially change the perspectives regarding the management and prognosis of refractory epilepsy [19]. In this way, this study aims to review the literature on CNB’s pharmacological properties, pharmacokinetics, efficacy, and safety. For a complete description of the methodology, read the Appendix A.

## 2. Historical Aspects of Cenobamate

Epilepsy is a neurological disorder characterized by recurrent and unprovoked seizures [20]. It is believed that excessive excitability in neural tissues can contribute to the abnormal electrical activities leading to epilepsy [21]. ASMs acting on voltage-gated sodium channels have been utilized for the pharmacologic management of epilepsy because these channels are essential for generating and conducting action potentials [22]. Also, several point mutations in voltage-gated sodium channels, which exhibit increased persistent sodium currents, have been identified in patients with epilepsy [23]. Some ASMs, such as phenytoin and lamotrigine, are known to affect persistent sodium currents [24].

In 1951, during rat studies to develop a new anxiolytic drug, meprobamate was observed to have antiseizure activity [25]. Ten years later, Frank Berger at Wallace Laboratories noted the remarkable efficacy of felbamate in controlling abnormal electrical activity in animal models of epilepsy (Figure 1) [26]. In 2008, Johnson & Johnson submitted a new application for carisbamate, which was approved by the U.S. Food and Drug Administration [27]. But, two years later, carisbamate was removed from the market due to insignificant superiority over a placebo in a randomized controlled trial [28].

Cenobamate (CNB), ([(R)-1-(2-chlorophenyl)-2-(2H-tetrazol-2-yl)ethyl], is a novel tetrazole alkyl carbamate derivative [29]. It showed antiseizure activity in the maximal electroshock test and prevented seizures induced by chemical convulsants such as pentylenetetrazol and picrotoxin [30]. Also, CNB was reported to be effective in two models of focal seizure, the hippocampal kindled rat and the mouse 6 Hz psychomotor seizure models [31]. Moreover, CNB has been reported to effectively and dose-dependently reduce the number and cumulative duration of spike-and-wave discharges characteristic of absence seizures in genetic absence epilepsy rats in the Strasbourg (GAERS) model [32]. For further assessment of the antiseizure potencies of alkyl-carbamates, read Table 1 [33].

CNB differs from other broad-spectrum ASMs because it has a sustained efficacy and potency in the 6 Hz test regardless of the stimulus intensity. Löscher et al., believed this significant efficacy predicted good outcomes in clinical trials [34]. In this way, the 6 Hz test should be studied as a therapy-resistant seizure model, and other ASMs should be assessed for a complete understanding of this model in DRE. Therefore, the significant efficacy of CNB in different models suggests that this drug may have a broad spectrum of activity [35].

The photosensitivity model has been established as a “proof-of-concept” study to achieve a reliable prediction of potential efficacy and chronic-use dosing of the early period of clinical trials with new ASMs. In this model, people with epilepsy are randomly distributed to take the tested drug or placebo as an adjunct single dose. An electroencephalogram (EEG) is used to observe abnormalities in the photoparoxysmal response [36]. Trenite et al., performed a single-blind non-randomized study with the photoparoxismal-EEG response (PPR) model in seven individuals with photosensitive epilepsy after oral doses of CNB of 100, 250, and 400 mg or placebo. A complete suppression of PPR response in photosensitive individuals at 250 and 400 mg single doses of CNB was observed. The subjects taking the CNB 100 mg dose had only partially suppressed PPR [37]. Thus, these results provide evidence of CNB’s potential efficacy in managing seizures in patients with epilepsy and support the clinical trials with this medication.

New methods to determine the plasma levels of CNB were developed to assess the pharmacokinetics of this drug in clinical trials. The first high-performance liquid chromatography-tandem mass spectrometry (LC-MS/MS) method was designed [38]. Carisbamate was used as the internal standard, and the preparation of plasma samples required the precipitation of proteins by acetonitrile. The calibration curve of this method was linear over a concentration range of 10–5000 ng/mL [39]. An achiral LC-MS/MS method was validated in heparinized plasma samples for pharmacokinetic studies. Phenacetin was used as an internal standard, and plasma samples with precipitation of proteins were analyzed. A third method of LC-MS/MS was developed for pharmacokinetic studies of CNB in plasma after administering a single-capsule formulation of 400 mg. The calibration curve range was 0.080–40.0 mg/L [40]. The fourth method to quantify CNB in human plasma samples was developed using ultra-high-performance liquid chromatography coupled with tandem mass spectrometry. The calibration curve range was 0.050–20.0 mg/L [41].

The first CNB trials revealed that more than twenty percent of people with epilepsy became seizure-free. Interestingly, this seizure freedom has never been reported in a placebo-controlled, double-blind trial of anticonvulsive drugs [42]. In this context, the FDA decision regarding CNB approval for marketing was remarkable. The advisory board recommended that additional randomized controlled phase 3 trials to investigate the efficacy of CNB were not necessary because of the impressive efficacy data in the phase 2 trials. Also, they replied that open-label safety data should be performed because of some rare cases of drug rash with eosinophilia and systemic symptoms (DRESS) in early trials [43,44].

The development of new routes of drug administration is important for the improvement of adherence. The administration of medications via enteral feeding tubes may be necessary for patients who cannot swallow safely, such as individuals with dysphagia due to cognitive impairment or physical disability. Ferrari et al., studied the recovery of CNB after the administration of a suspension prepared from filmcoated tablets via ex vivo nasogastric and gastrostomy feeding tubes. The authors observed that the mean percentage of recovery from CNB was within the predetermined acceptable range (90.0–110.0%), which can suggest no adhesion or adsorption of CNB to enteral feeding tubes [45]. Thus, enteral feeding tubes may be suitable for the administration of CNB.

## 3. Pharmacology and Mechanism of Action

CNB’s mechanism of action has yet to be completely understood (Table 2). Interestingly, CNB was discovered purely by phenotype-based screening, and its presumed dual mechanism of action was only described years after the first studies [46]. CNB can reduce repetitive neuronal firing by inhibiting voltage-gated sodium currents. It may enhance the fast and slow inactivation of sodium channels and potently inhibit the non-inactivating persistent component of the sodium channel current, which has already been observed with other ASMs [47]. Noteworthily, CNB had little effect on the peak component of transient sodium currents induced by brief depolarizing step pulses. But, CNB strongly inhibited the noninactivating persistent component of sodium currents [48]. Therefore, CNB may modify excitability in principal neurons without compromising inhibitory interneurons [49]. Also, CNB was revealed to be a positive allosteric modulator of the γ-aminobutyric acid (GABA) ion channel. This effect was similar for all tested GABAA receptors containing six different alpha subunits (α1β2γ2 or α2-6β3γ2).

Nakamura et al., studied the effects of CNB in rat hippocampal CA3 neurons. They observed that CNB had little effect on the peak component of transient sodium current induced by brief depolarizing step pulses. Still, CNB potently inhibited the non-inactivating persistent component of sodium currents. Also, it inhibited the sodium currents evoked by slow voltage-ramp stimuli [48]. Noteworthily, the effect of CNB in sodium current in hippocampal rat neurons was concentration-dependent [43].

Sharma et al., assessed the effects of CNB on GABAergic neurotransmission, specifically its effects on GABAA receptors mediating inhibitory postsynaptic currents and tonic conductance in rodent hippocampal neurons. The authors found that CNB is a positive allosteric modulator of high-affinity GABAA receptors, activated by GABA at a site independent of the benzodiazepine binding site, and efficiently enhances tonic conductance inhibition in hippocampal neurons [50]. CNB may resemble barbiturate action because of increased tonic and phasic inhibition through GABAA receptor activation [51]. It is worth mentioning that these mechanisms were already observed in animal studies with neurosteroids [52]. Also, the effect on both phases of GABAA receptor activation could partially explain the efficacy of CNB in managing status epilepticus [53].

The CNB terminal half-life of 50 to 60 h allows this drug to be taken once a day. Noteworthily, this terminal half-life increases with increasing doses of CNB from 30 h (CNB 10 mg) to 76 h (CNB 750 mg). The area under the plasma concentration versus time curve (AUC) increases more than proportionally after the administration of single doses of CNB ranging from 5 to 750 mg. However, after multiple doses of CNB at the steady state, AUC increases linearly with increasing doses within the 50–500 mg/day dose range. Plasma CNB concentrations are steady after approximately two weeks of once-daily dosing. The tablets should be swallowed whole and not crushed or chewed. This drug is extensively metabolized via glucuronidation and oxidation, so drug interactions can occur (Table 3) [32,54,55,56].

CNB pharmacokinetics have been reported to be consistent regarding gender, race, and age. Patients with mild-to-moderate (Clcr 30 to 90 mL/min) and severe (Clcr 30 mL/min) renal impairment and those with mild-to-moderate hepatic impairment should be treated with caution and reduced dose. There are no data regarding CNB’s pharmacokinetics in individuals with end-stage renal disease (Clcr < 15 mL/min) undergoing hemodialysis and those with severe hepatic impairment.

Vernillet et al., studied the mass balance and the metabolic profiling of CNB in humans. Eight CNB metabolites (M1, M2a, M2b, M3, M5, M6, M7, and M11) were identified across plasma, urine, and feces. CNB was the main plasma radioactive component, and M1 was the only metabolite detected in plasma (>98% and <2% total radioactivity AUC, respectively). All detected metabolites were found in urine; unchanged CNB accounted for approximately six percent. CNB metabolites appeared to be formed slowly [40]. Greene et al., assessed the effect of CNB on the single-dose pharmacokinetics of multiple cytochrome P450 probes in healthy subjects. They observed that CNB induces CYP2B6 activity, exhibits a dose-dependent induction of CYP3A4/5 activity, inhibits CYP2C19 activity, and has a negligible effect on CYP2C9 activity [57].

We calculated the chemical and pharmacological properties of CNB using the SwissADME tool (Figure 2). These properties can help identify compounds suitable for oral use [58]. All the parameters analyzed were within the normal range, except for an insaturation slightly higher than those desired for oral molecules, which can reduce oral bioavailability. Refer to Table 2 for a complete description of the physicochemical descriptors and pharmacokinetic characteristics of CNB. Odi et al., assessed the physicochemical and biopharmaceutic properties of marketed ASMs. CNB has the highest polar surface area value among third-generation ASMs [59].

## 4. Clinical Trials

There are 18 clinical trials assessing CNB’s efficacy and therapeutic management registered in the ClinicalTrials.gov database (Figure 3) (Table 4). Enrollment involved a total of 3964 individuals. Eleven studies evaluated the efficacy of CNB in the management of focal epilepsy. NCT03678753 and NCT03961568 are essential clinical trials because they studied CNB efficacy in primary generalized epilepsy. There are at least five studies awaiting the release of tabular results.

The pivotal clinical trials were Study CO13 (NCT01397968, Chung et al.) [60], Study CO17 (NCT01866111, Krauss et al.) [61], and Study CO21 (NCT02535091, Sperling et al.) [62] (Table 5). Compared to placebo, these studies revealed a significant efficacy of CNB in median percentage seizure reduction from baseline and seizure freedom during the maintenance phase.

We also revised the literature and included the reports already published with cenobamate (YKP3089) (Table 6).

## 5. Discussion

### 5.1. Efficacy of Cenobamate

Lattanzi et al., systematically reviewed the results from the studies CO13 and CO17. The pooled estimated risk ratio to achieve seizure freedom for the CNB group compared to the placebo was 3.71 (95%, CI: 1.93–7.14; *p* < 0.001). Also, seizure frequency reduction by at least 50% occurred during the maintenance phase in 50.1% of the patients randomized to CNB and 23.5% of the placebo-treated participants (RR 2.18, 95% CI 1.67–2.85; *p* < 0.001). Therefore, adjunctive CNB therapy in adult patients with uncontrolled focal-onset seizures is associated with a greater reduction in seizure frequency than placebo [93]. Another meta-analytic study by Zhang et al., with the studies CO13 and CO17 found similar results [94].

Cutillo et al., studied the literature about ASMs and their efficacy for controlling focal to bilateral tonic-clonic seizures. One of the ASMs studied was CNB, which showed significant efficacy compared to placebo (18–59% efficacy above placebo). These results are important because focal to bilateral tonic-clonic seizures are recognized as one of the major risk factors for sudden unexpected death in epilepsy (SUDEP). Therefore, decreasing the incidence of this specific type of focal epilepsy can probably lead to decreased overall mortality by SUDEP [95].

A network meta-analysis among the third-generation ASMs was performed [96]. CNB was associated with a higher rate of ≥ 50% seizure frequency reduction than brivaracetam [odds ratio (OR) 2.02, 95% confidence interval (CI) 1.11–3.66], eslicarbazepine (OR 1.93, 95% CI 1.07–3.48), lacosamide (OR 1.86, 95% CI 1.04–3.32), and perampanel (OR 2.07, 95% CI 1.16–3.70). However, no statistically significant trends favored CNB over the other ASMs for seizure freedom outcomes. In this way, CNB was the most effective ASM, but brivaracetam and lacosamide were the ASMs with fewer side effects [97].

Privitera et al., performed a network meta-analysis with CNB and seven other ASMs (brivaracetam, eslicarbazepine acetate, lacosamide, and perampanel, lamotrigine, levetiracetam, and topiramate). The placebo-adjusted 50% responder rate of CNB was superior (OR 4.200; 95% CI 2.279, 7.742) to all seven assessed ASMs (OR 2.202 95% CI 1.915, 2.532; *p* = 0.044). Also, there was no increasing percentage of treatment discontinuation by treatment-emergent adverse events of CNB compared to other ASMs [98].

### 5.2. Cost-Effectiveness of Cenobamate

Despite greater efficacy, CNB is still infrequently prescribed. Klein et al., reported that after two years of United States market entry, only less than five percent of adults with focal DRE are treated with CNB. They explained that this possibly occurred due to restrictions to access created by the healthcare system, insufficient post-launch information about efficacy and safety, and limited knowledge about this drug. Also, Klein et al., compared the costs among CNB and other new ASMs approved since 2009, and the cost was similar in the United States and Germany [99].

Flint et al., developed a mapping algorithm to predict SF-6D values in adults with focal-onset seizures for use in economic evaluations of CNB [100]. This preference-based measure of health-related quality of life can assess six dimensions of health status, including physical functioning, role limitation, social functioning, pain, mental functioning, and vitality [101]. The authors found that the mapping algorithm proposed may predict SF-6D values from clinical outcomes in people with epilepsy [100]. Therefore, researchers can use outcome data from clinical trials with CNB to facilitate cost–utility analysis.

A Markov model simulation of DRE in Spain was performed to analyze the cost-effectiveness of CNB with other ASMs (brivaracetam, eslicarbazepine acetate, lacosamide, and perampanel). The authors found that CNB’s daily economically justifiable price of EUR 7.30 is cost-effective for a threshold of EUR 21,000/quality-adjusted life-years. Thus, CNB can produce more health per invested euro. Calleja et al., suggested that CNB therapy can produce an incremental clinical benefit over third-generation ASMs [102].

Laskier et al., estimated the cost-effectiveness of add-on CNB in the UK when used to treat drug-resistant focal seizures in adults. They found that CNB led to cost savings of GBP 51,967 (compared to brivaracetam), GBP 21,080 (compared to eslicarbazepine), GBP 33,619 (compared to lacosamide), and GBP 28,296 (compared to perampanel). They also found an increased cost per quality-adjusted life-year of 1.047 (compared to brivaracetam), 0.598 (compared to eslicarbazepine), 0.776 (compared to lacosamide), and 0.703 (compared to perampanel) per individual over a lifetime time horizon. Therefore, CNB is less costly and more effective when compared to brivaracetam, eslicarbazepine acetate, lacosamide, and perampanel [103].

Villanueva et al., assessed the clinical benefit regarding the number needed to treat (NNT), efficiency, and cost per NNT (CNT) associated with CNB versus third-generation ASMs used in Spain for the adjunctive treatment of focal-onset seizures in patients with DRE. CNB was the ASM associated with the lowest values of NNT at all doses for both fifty-percent responder rate and seizure freedom compared with the alternatives. Also, CNB was the ASM associated with the lowest CNT values at the daily defined dose (DDD) and minimum lacosamide and maximum eslicarbazepine acetate dose for fifty-percent responder rate. Moreover, the maximum dose of CNB was associated with the lowest CNT value at DDD and the minimum dose of lacosamide for seizure freedom [104].

### 5.3. Neuroprotective Potential of Cenobamate

In 2010 (Tenth Eilat Conference on New Antiepileptic Drugs) and 2013 (Eleventh Eilat Conference on New Antiepileptic Drugs), Bialer et al., reported that CNB showed neuroprotective abilities in the hypoxia-induced lethality mice model [105]. In this context, Wiciński et al., performed a review of the literature about the neuroprotective effects of CNB. They believe that the pharmacodynamics of CNB may confer excellent neuroprotective properties [106].

There are five main pathways to explain the neuroprotective effect of CNB. First, CNB may inhibit depolarization and signal propagation by blocking voltage-gated sodium channels [107]. This can be further enhanced by positive allosteric modulation of GABAA receptors, leading to chloride efflux and hyperpolarizing the membrane [108]. Third, CNB, through GABAA receptors, can activate the PI3K/Akt pathway, which, when phosphorylated, can modulate gene transcription and decrease protein degradation, promoting cell survival and self-renewal [109]. Fourth, the inhibition of depolarization can prevent voltage-gated calcium activation, which does not allow glutamate release in the synaptic cleft and, consequently, excitotoxicity [110]. Fifth, the widespread blockade of persistent sodium currents through voltage-gated sodium channels prevents the increased depolarization and activity of central nervous system neurons [106].

### 5.4. Cognition and Cenobamate

Sodium channels play an essential role in dendritic sodium spike generation, related to the Hebbian long-term potentiation of excitatory synaptic transmission and cognitive function [111]. Also, dysfunctional GABAergic activity in the pre-frontal cortex may lead to working memory and cognitive impairments [112].

A low rate of cognitive and psychiatric treatment-emergent adverse events was observed during adjunctive CNB treatment in clinical trials. Song et al., studied the effects of CNB on cognitive behaviors and hippocampal long-term potentiation in mice. The authors showed that CNB influenced novel object recognition, object location memory, and Morris water maze performance in mice. Also, CNB suppressed hippocampal excitatory synaptic transmission by reducing the excitability of Schaffer collaterals and interfered with the induction of long-term potentiation [113]. Therefore, CNB can potentially affect cognitive function in animal models of epilepsy, but there was no report in the clinical trials.

### 5.5. Electrocardiographic Abnormalities Associated with Cenobamate

Patients treated with CNB may experience a shortening of the QT interval, and this new agent is contraindicated in patients with familial short QT syndrome because of the increased risk of ventricular dysrhythmias and sudden death [75]. A dose-dependent QT-interval shortening was observed with CNB. In studies with healthy subjects, the CNB dose of 200 mg/day was associated with QT-interval shortening of more than 20 ms in 33% of the individuals. However, when the dose of CNB was increased to 500 mg/day, 66% of the subjects developed electrocardiographic abnormalities [114]. Concurrent use with other drugs that shorten the QT interval (for example, lamotrigine and rufinamide) should be closely monitored.

Darpo et al., performed a QT study to assess the effects of therapeutic and supratherapeutic CNB doses (maximum recommended dose, 400 mg/day) on correct QT interval (QTc) in healthy adults. The authors found that CNB had no clinically relevant effects on heart rate or electrocardiographic parameters and no QTc-prolonging effect at therapeutic/supratherapeutic doses [75].

### 5.6. Hepatotoxicity Secondary to Cenobamate

CNB was associated with a low-to-moderate rate of serum aminotransferase elevations during therapy in less than five percent of individuals. The hepatotoxic side effect was usually observed with hypersensitivity reactions such as drug reactions with eosinophilia and systemic symptoms (DRESS) [115].

In a study with 953 CNB users, three presented DRESS and abnormal serum liver enzymes within six weeks of CNB therapy onset [116]. However, there were no reports of serum aminotransferase elevations in a larger sample with an adequate titration period [62].

Felbamate is a carbamate anticonvulsant, as is CNB. Felbamate has also already been associated with hepatotoxicity. The black box warning about felbamate has a warning regarding the risk for acute liver failure and that felbamate should only be started in patients with normal liver function [117]. Thus, hepatotoxic side effects in the carbamate class can occur. However, there is only a slight association between hepatotoxicity and CNB.

Interestingly, the CNB titration protocol should be cautiously followed because it shows fewer total side effects, including DRESS. Krauss et al., revealed that DRESS syndrome was seen only with a relatively rapid initial titration period [61]. The management of CNB-induced DRESS is discontinuing CNB and prescribing corticosteroid therapy and antihistamines [60].

Another important fact regarding CNB prescription is the introduction of this drug in patients already taking another voltage-gated sodium channel blocker. Therefore, a slower titration of CNB is advised, especially in individuals with other voltage-gated sodium channel blocker therapies, as well as stepwise tapering of the previously ineffective agent when starting CNB. This may increase tolerability and reduce the risk of treatment failure due to adverse events [118].

### 5.7. Cenobamate-Induced Movement Disorders

Sáenz-Farret et al., reviewed the literature on movement disorders secondary to antiseizure medications. They found one manuscript of CNZ-induced tremor [119]. In a randomized controlled trial of adjunctive CNB in patients with uncontrolled focal seizures, tremor was reported by seven (6.2%) CNB users and by three (2.8%) individuals in the placebo group. A detailed description of phenomenology or localization was not provided [60]. In this context, felbamate was reported with akathisia, chorea, and dystonia [120]. It is worth mentioning that due to structural similarities between these two medications, these abnormal movements could also be seen with CNB.

A pathophysiological explanation for the development of abnormal movements can be indirectly related to GABAergic neurotransmission. Therefore, case reports of dystonia and tremor will probably be observed in the future, as reported with other ASMs, such as pregabalin [121] and valproate [122].

### 5.8. Pregnancy and Lactation

There are no data regarding the developmental risk associated with using CNB in pregnant women [123]. Also, data are unavailable on the presence of CNB in human milk, its effects on breastfed infants, or its effects on milk production [124]. However, the administration of CNB in animal studies during pregnancy or throughout lactation significantly increased the risk of developmental abnormalities, such as increased embryofetal mortality, decreased fetal and offspring body weights, and neurobehavioral and reproductive impairment in offspring [53]. In a systematic review of breastfeeding while on treatment with antiseizure medications, the authors did not find reports of CNB levels in breastmilk or breastfed infants [125].

CNB may decrease the plasma concentration of oral contraceptives. Therefore, women of reproductive age using oral contraceptives should use dual protection, which can be accomplished by consistently using male/female condoms [54].

### 5.9. Potential for Abuse and Suicidality Risk

CNB is scheduled for class 5 by the Drug Enforcement Administration (DEA). Class 5 is characterized by medications with the least potential for abuse of controlled substances. The ASMs developed before 2006 were standardly classified as class 5 [126]. This scheduling is based on the adverse effects observed in clinical studies with debatable applicability to patients with epilepsy. DEA scheduling has significant drawbacks, such as apprehension by patients in using a medication related to addiction, low storage of the medication by pharmacies, and patients with epilepsy being unable to refill prescriptions until they have used up their existing pills. These factors probably affected access to CNB and led some individuals to discontinue or even avoid this medication [127]. Noteworthily, the discontinuation of CNB should be performed gradually to reduce the risk of withdrawal syndrome and increased seizure frequency.

The allosteric modulation of GABAA receptors by CNB occurs at a site independent of the benzodiazepine binding site. So, the risk of inducing dependence, withdrawal symptoms, and tolerance is lower with CNB than with benzodiazepines [50]. CNB’s potential for abuse and dependence was observed in animal models, which has been further assessed with studies using alprazolam and placebo as comparators. In this single-dose, randomized, double-blind, placebo-controlled crossover study (YKP3089C024), CNB’s abuse potential profile was significantly lower than that of alprazolam. Euphoric mood was observed in 2% of placebo, 19% of alprazolam 1.5 mg, 17.4% of alprazolam 3 mg, 0% of CNB 200 mg, and 18% of CNB 400 mg patients [32].

Most ASMs have a class of warning by the U.S. Food and Drug Administration for increased suicidality risk (suicidal ideation and behavior). This is based on a retrospective meta-analysis of randomized clinical trials of eleven ASMs approved between 1990 and 2007, comparing suicidality in patients treated with ASMs and placebo [126]. Klein et al., recently reviewed all randomized, placebo-controlled phase 2 and 3 studies of new ASMs to compare suicidality rates between patients treated with ASMs and with placebo to determine whether these agents are associated with increased risk for suicidality. There was no evidence of increased risk of suicidal ideation (ASMs and placebo overall risk ratio, 0.75; 95% CI, 0.35–1.60) or attempt (risk ratio, 0.75; 95% CI, 0.30–1.87) overall or for any individual drug. The authors found no current evidence that CNB and other ASMs (eslicarbazepine, perampanel, brivaracetam, and cannabidiol) increase suicidality in epilepsy and merit a suicidality class warning [128].

## 6. Expert Recommendations

For instance, it is evident that, similarly to lamotrigine, the risk of allergic and immunologic adverse events can be markedly reduced by a very cautious and slow titration [129]. A more gradual titration schedule was developed to reduce the incidence of side effects associated with CNB. This schedule was investigated in Study CO21. CNB dosing approved by the U.S. Food and Drug Administration protocol is shown below (Table 7).

Patients should be advised against engaging in activities requiring mental alertness, such as operating vehicles or machinery, until the response to CNB has been determined during the titration phase. Concomitant use with other central nervous system depressants or alcoholic beverages may have additive sedative side effects, which should be advised. Also, patients should be monitored for the emergence or worsening of depression and/or any unusual changes in mood or behavior, as well as suicidal thoughts or behavior [130]. Moreover, it is worth remembering that individuals with DRE usually take many other ASMs, and there is a significant risk of interactions (Figure 4).

To prevent pharmacodynamic adverse events, decreasing the baseline medication as early as possible during the titration of CNB is helpful so that such side effects due to interactions can be avoided and individual adherence is improved. It may also be helpful to measure the plasma levels of the concomitant medication to assess the individual course closely.

Steinhoff et al., provided practical guidance for managing adults receiving adjunctive CNB to treat focal epilepsy [131]. A summary of the main takeaway points is provided in Table 8.

## 7. Future Perspectives

DRE commonly requires polytherapy. But, clinicians have scarce guidance on how to approach polytherapy for epilepsy. So, a systematic evaluation of the possibilities for polytherapy in the treatment of uncontrolled seizures should be performed. CNB revealed significant efficacy for these refractory cases. Therefore, studies with CNB as monotherapy and polytherapy should be continuously performed.

Another important aspect to evaluate in future studies is the CNB spectrum of action. CNB should be assessed in generalized epilepsy, specific epilepsy syndromes, and even other comorbidities. Since the other alkyl carbamate derivatives demonstrated efficacy for managing anxiety and neuropathic pain, it is possible that CNB could also be efficacious in treating these conditions. Moreover, special patient groups such as infants, children, elderly, and patients with epilepsy and other comorbid conditions should be included in the coming clinical trials. CNB non-linear kinetics may cause more-than-proportional drug concentrations at high doses with frequent possible intolerable neurological adverse effects. Further studies are required to better-characterize pharmacokinetic and pharmacodynamics interactions with co-administered medications and the relationships between plasma CNB concentrations and clinical effects.

Continuing safety data with open-label trials are recommended through drug surveillance activities during routine clinical use. The need for a slow titration is still a significant drawback of CNB for the management of patients with status epilepticus, episodes of cluster seizures, and acute repetitive seizures. Therefore, further studies with fast titration or even developing an acute formulation with similar features should be developed.

Network meta-analysis showed significant superiority of CNB when compared with other third-generation ASMs. Head-to-head trials with different ASMs should be performed to support these findings. Moreover, the clinical databases with CNB trials should be assessed for highly efficacious combinations of ASMs. It would be important to highlight some clinical findings or even genetic features leading to good outcomes with CNB therapy.

In Study CO17, people with epilepsy taking only one ASM were included. In this way, some specialists believe that a higher proportion of less-severe epilepsy was included in this pivotal trial. Another discussed point in CO13 and CO17 was the relatively short titration and maintenance phases. The individuals with epilepsy in these trials reached a steady-state concentration within ten days. Thus, the interpretation of short-term adjunctive trials is challenging and sensitivity analyses for differences in baseline seizure frequency would have been beneficial in these trials.

## 8. Conclusions

CNB is a highly effective drug in managing focal onset seizures, with more than twenty percent of individuals with DRE achieving seizure freedom. This finding is remarkable for the antiseizure medication literature and combines with the approval for marketing after the impressive efficacy data of the phase 2 trials. The CNB mechanism of action is still poorly understood, but it is associated with transient and persistent sodium currents and GABAA receptors. In animal studies, CNB showed sustained efficacy and potency in the 6 Hz test regardless of the stimulus intensity. CNB was revealed to be the most cost-effective drug among different third-generation ASMs. However, there are still concerns regarding the potential for abuse and suicidality risk, which future studies should clearly assess, and protocols should be changed. The major drawback of CNB therapy is the slow and complex titration and maintenance phases preventing the wide use of this new agent in clinical practice.

## Figures and Tables

**Figure 1 medicina-59-01389-f001:**
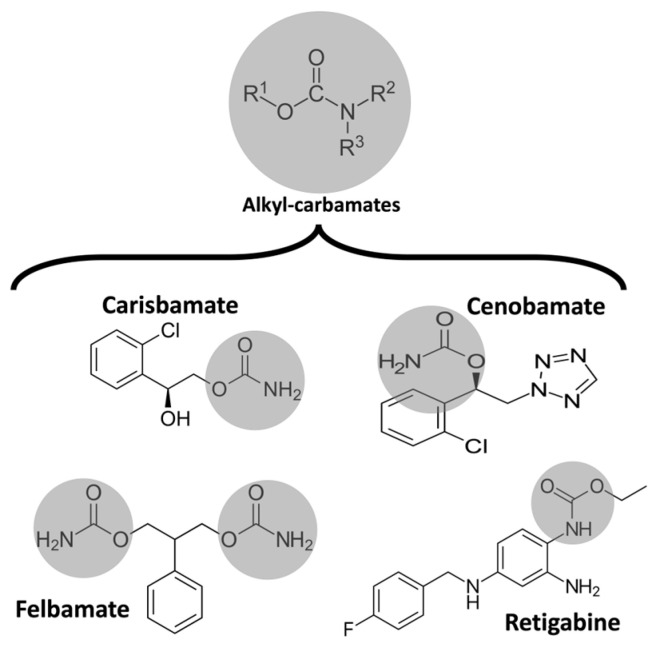
Chemical structure of some alkyl-carbamates with antiseizure activity. Carisbamate, cenobamate, felbamate, and retigabine (ezogabine). Note that felbamate is a dicarbamate. The other drugs are monocarbamates.

**Figure 2 medicina-59-01389-f002:**
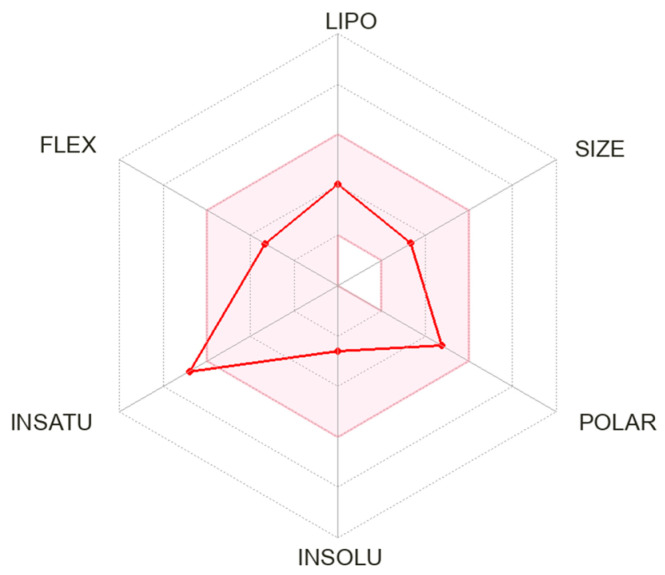
Physicochemical properties of cenobamate. The pink area represents the optimal range for each property. Abbreviation: LIPO: lipophilicity; FLEX: flexibility; INSATU: saturation; INSOLU: solubility.

**Figure 3 medicina-59-01389-f003:**
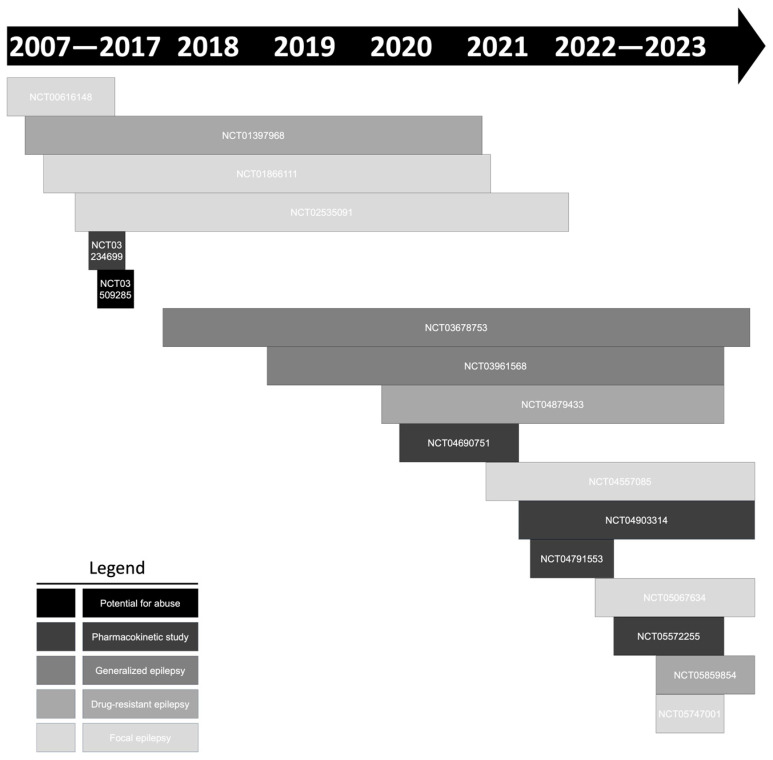
Clinical trials with cenobamate (YKP3089) registered in the ClinicalTrials.gov database.

**Figure 4 medicina-59-01389-f004:**
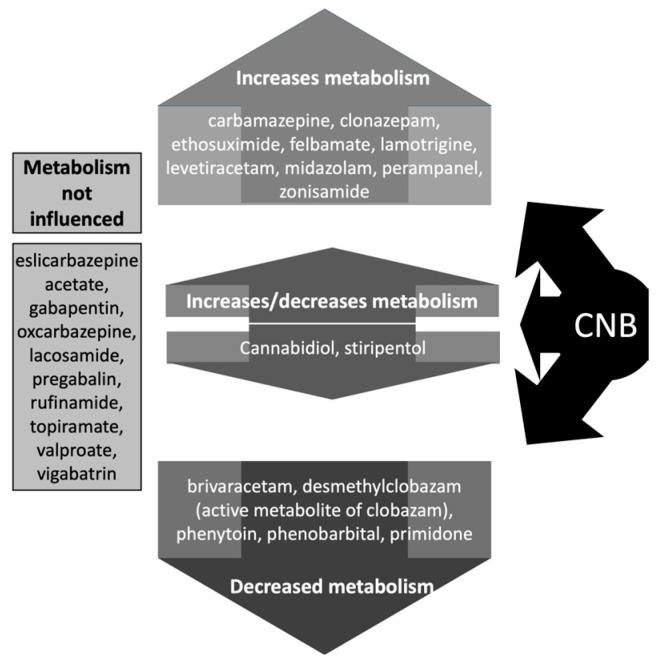
Effects of cenobamate (CNB) on drug levels of antiseizure medications (ASMs). For ASMs whose metabolism can be increased by CNB, consider increasing the dose of the ASM. For ASMs whose metabolism can be decreased by CNB, consider reducing the dose of the ASM.

**Table 1 medicina-59-01389-t001:** Antiseizure potencies of alkyl-carbamates in mouse and rat models by Löscher et al. [33] adapted by Rissardo et al. ^a,b^.

Alkyl-Carbamate		Carisbamate	Cenobamate	Felbamate	Retigabine
Mechanism of Action		AMPA, NMDA, Transient Sodium Currents, VGCC	GABAA Receptors and Persistent Sodium Currents	GABAA and NMDA Receptors, Transient Sodium Currents, VGCC	Voltage-Gated Potassium Channels, GABAA Receptors
MES	Mice	7.9	9.8	35.5	9.3
	Rats	4.4	2.9	35	5.1
PTZ	Mice	20.4	28.5	126	149
	Rats	NA	NA	>250	195
6 Hz (mice)	22 mA	20.7	11	13.1	NA
	32 mA	21.4	17.9	69.5	26
	44 mA	27.6	16.5	241	33
Rotarod test	Mice	46	58	220	20.5
	Rats	39.5	38.9	>500	10
Kindled seizures ^c^		NA	16.4	296	3.2

Abbreviations: AMPA: α-amino-3-hydroxy-5-methyl-4-isoxazolepropionic acid; GABA: γ-Aminobutyric acid; MES: maximal electroshock seizure; NA: not available/not reported; NMDA: N-methyl-D-aspartate; PTZ: pentylenetetrazole; VGCC: voltage-gated calcium channels. ^a^ Results are related to ED50 (mg/kg i.p.) at the time of peak effect. ^b^ Potency varies with the mouse strain used. Also, ED50s are higher for focal seizures. ^c^ Amygdala/hippocampal kindled seizures.

**Table 2 medicina-59-01389-t002:** Pharmacological properties of cenobamate ^a^.

Dosage forms and strengths (mg)		12.5, 25, 50, 100, 150, 200
Bioavailability		88%, not influenced by high-fat meal
Peak plasma time:		1–4 h
Volume distribution		40–50 L
Plasma-protein binding		60%, primarily to albumin
Metabolism		Glucuronidation (UGT2B7 > UGT2B4) Oxidation (CYP2E1, CYP2A6, CYP2B6 > CYP2C19, CYP3A4/5)
Elimination half-life (hours)	Human	50–60
	Rat	2.9
	Mice	NA
Oral clearance		0.45–9.63 L/hr
Excretion		Renal 87.8% (6% unmetabolized) and feces 5.2% (<1% unmetabolized)
Physicochemical properties extracted from SwissADME ^b^		
Formula		C_10_H_10_ClN_5_O_2_
Molecular weight (150–500 g/mol)		267.67 g/mol
Fraction Csp3 (0.25–1.0)		0.20
Num. rotatable bonds (0–9)		5
Num. H-bond acceptors (≤10)		5
Num. H-bond donors (≤5)		1
Topological polar surface area (20–130 Å^2^)		95.92 Å^2^
Lipophilicity		
Consensus Log Po/w (0.7–5.0)		0.95
Water solubility		
Log S (ESOL) (−6–0)		−2.59
Class		Soluble
Pharmacokinetics		
Gastrointestinal absorption		High
Blood–brain barrier permeant		No
Druglikeness		
Bioavailability score		0.55
Medicinal chemistry		
Synthetic accessibility [1(very easy)–10 (very difficult)]		3.05

Abbreviations: NA, not available/not reported. ^a^
https://www.xcopri.com (accessed on 30 May 2023). ^b^ Normal range provided is desirable for oral drugs. Consensus Log Po/w is the average of iLOGP, XLOGP3, WLOGP, MLOGP, and SILICOS-IT.

**Table 3 medicina-59-01389-t003:** Drug–drug interactions between cenobamate and other medications by Barbieri et al. [32], Smith et al. [55], and Villani et al. [56], adapted by Rissardo et al.

Medication	Effect of CNB on Drug/Substrate	Mechanism	Recommendation
CBZ	Decrease of 24% in plasma CBZ levels	CYP3A4 induction	Monitor plasma CBZ level and increase CBZ dose as needed
CLB	Increase in plasma N-desmethylclobazam (active metabolite of CLB) levels	CYP2C19 inhibition	Monitor plasma N-desmethylclobazam levels and decrease CLB dose as needed
LTG	Decreases of 21% (CNB 100 mg/day), 35% (CNB 200 mg/day), and 52% (CNB 400 mg/day) in plasma LTG levels	Induction of UDPGT	Monitor plasma LTG levels and increase LTG dose as needed
PB	Increase in PB AUC by 37%	CYP2C19 inhibition	Monitor plasma PB levels and decrease PB as needed
PHT	Increase in PHT AUC by 84%	CYP2C19 inhibition	Monitor plasma PHT levels. Gradually decrease PHT dose by up to 50% during CNB titration
OCP	Decrease in plasma concentrations of OCPs	CYP3A4 induction	Use additional or alternative non-hormonal birth control methods
CYP2B6 substrates	Decrease in plasma concentrations of CYP2B6 substrates, e.g., decrease in plasma bupropion levels by 39%	CYP2B6 induction	Increase the dosage of CYP2B6 substrates as needed
CYP2C19 substrates	Increase in plasma concentrations of CYP2C19 substrates, e.g., increase in plasma omeprazole levels by 107%	CYP2C19 inhibition	Monitor plasma concentration or response to CYP2C19 substrates and decrease the dose of CYP2C19 substrates as needed
CYP3A4 substrates	Decrease in plasma concentrations of CYP3A4 substrates, e.g., decrease in plasma midazolam levels by 27% (CNB 100 mg/day) to 72% (CNB 200 mg/day)	CYP3A4 induction	Increase the dosage of CYP3A4 substrates as needed
Drug-induced QT interval shortening	Additive effect on QT interval shortening	Variable	Drugs associated with QT-interval shortening should be cautiously prescribed when in combination with CNB
Drug-induced CNS side effects	Additive effect of CNS depressants	Variable	CNS depressants should be cautiously prescribed when in combination with CNB

Abbreviations: AUC: area under curve; CBZ: carbamazepine; CLB: clobazam; CNB: cenobamate; CNS: central nervous system; CYP: cytochromes P450; LTG: lamotrigine; PB: phenobarbital; OCP: oral contraceptive; PHT: phenytoin; UDPGT: uridine 5′-diphospho-glucuronosyl transferase.

**Table 4 medicina-59-01389-t004:** Clinical trials with cenobamate (YKP3089) registered in the ClinicalTrials.gov database.

Identifier	Study Start to Completion	Condition	Intervention	N Enrolled	Comment
NCT04513860	NA	Focal epilepsy	CNB	NA	The objective of this expanded access program is to continue providing treatment with CNB to patients with focal epilepsy that were enrolled in the SK Life Science Inc clinical trials
NCT00616148	Aug 2007–Jan 2010	Focal epilepsy	CNB, placebo	11	PPR study
NCT01397968	Jul 2011–Jan 2021	Focal epilepsy	CNB, placebo	222	Efficacy of CNB in DRE
NCT01866111	Jul 2013–Oct 2021	Focal epilepsy	CNB, placebo	437	Effective dose range of CNB as adjunctive therapy
NCT02535091	Aug 2016–Feb 2022	Focal epilepsy	CNB	1345	Effective dose range of CNB as adjunctive therapy
NCT03234699	Feb 2017–Jul 2017	Healthy	CNB, midazolam, warfarin, omeprazole, bupropion	24	Investigate the influence of CNB on the activity of CYP3A4/5, CYP2B6, CYP2C19, and CYP2C9
NCT03509285	Mar 2017–Dec 2017	Healthy	CNB, alprazolam	53	Evaluate the abuse liability potential of CNB in recreational drug users with sedative drug use experience
NCT03678753	Sep 2018–Jul 2024	Primary generalized epilepsy	CNB, placebo	170	Safety and effectiveness of CNB on primary generalized tonic-clonic seizures
NCT03961568	Aug 2019–May 2023	Primary generalized epilepsy	CNB	130	Long-term safety of CNB adjunctive therapy in subjects with primary generalized tonic-clonic seizures
NCT04879433	Jun 2020–Nov 2023	Focal epilepsy	CNB	100	Efficacy, safety, and tolerability of CNB as adjunctive treatment of DRE
NCT04690751	Dec 2020–May 2021	Healthy	CNB	28	Pharmacokinetics of CNB
NCT04557085	Mar 2021–Oct 2024	Focal epilepsy	CNB, placebo	540	Efficacy and safety of 100, 200, and 400 mg/day of CNB as adjunctive therapy in focal epilepsy
NCT04903314	May 2021–Oct 2024	Focal epilepsy	CNB	24	Pharmacokinetics of CNB in pediatric subjects
NCT04791553	Jun 2021–Nov 2022	NA	CNB	16	Effect of severe hepatic impairment on the pharmacokinetics of CNB
NCT05067634	Jan 2022–Jul 2026	Focal epilepsy	CNB	140	Safety and tolerability of CNB in pediatric subjects with focal epilepsy
NCT05572255	Sep 2022–Jan 2023	Healthy	CNB	24	Pharmacokinetics of CNB
NCT05859854	Jan 2023–Sep 2024	Focal epilepsy	CNB	200	Efficacy of CNB in DRE
NCT05747001	Jan 2023–Apr 2023	Focal epilepsy	CNB	500	Effectiveness and tolerability of CNB from real-world data collected in patients who participated in the early access program

Abbreviations: CNB: cenobamate; CYP: cytochromes P450; DRE: drug-resistant epilepsy; PPR: photoparoxismal-electroencephalogram response (PPR) model.

**Table 5 medicina-59-01389-t005:** Clinical studies CO13, CO17, and CO21 of adjunctive cenobamate efficacy and safety.

Reference		Study CO13, NCT01397968, Chung et al. [60]	Study CO17, NCT01866111, Krauss et al. [61]	Study CO21, NCT02535091, Sperling et al. [62]
Type of study		Phase II, R, DB, followed by OLE	Phase II, R, DB, DR, followed by OLE	Phase III, open-label
Seizure type		Focal, uncontrolled ^a^	Focal, uncontrolled ^b^	Focal, uncontrolled ^b^
CNB starting dose mg/day		50	50 ^c^	12.5
Titration schedule		Increase by 50 mg every two weeks	Increase by 50 mg every week up to 200 mg, then 100 mg/week thereafter ^c^	Increase to 25 mg for weeks 3 and 4, 50 mg for weeks 5 and 6, and then by 50 mg every 2 weeks thereafter
Titration phase, weeks		6	6	12
CNB target dose for maintenance phase mg/day (N of participants)		200 (n = 113)	100 (n = 108); 200 (n = 110); 400 (n = 111)	200, could be increased to a maximum dose of 400 (n = 1339)
Maintenance phase, weeks		6	12	≥40
Compared group		Placebo (n = 109)	Placebo (n = 108)	NA
Inclusion criteria	Common	1. Taking 1–3 concomitant ASMs at stable doses; 2. EEG confirming the diagnosis of focal epilepsy; 3. prior neuroimaging		
	Specific	4. Adults 18–65 years old 5. ≥3 focal seizures per month (baseline period 8 weeks) 6. No consecutive 21-day seizure-free interval	4. Adults 18–70 years old 5. ≥3 focal seizures per month (baseline period 8 weeks), with ≥8 focal seizures during baseline 6. No consecutive 25-day seizure-free interval	4. Adults 18–70 years old
Exclusion criteria	Common	1. Taking FBM for <18 continuous months; 2. history of status epilepticus, alcoholism, drug abuse, or psychiatric illness; 3. taking VGB within the past year		
	Specific	4. Taking intermittent rescue benzodiazepines more than once per month within the past month 5. Taking PHT or PB 6. History of >2 allergic reactions to prior ASMs 7. History of 1 serious hypersensitivity reaction	4. Taking intermittent rescue benzodiazepines more than once per month within the past month 5. Taking diazepam, PHT, or PB 6. History of a serious drug-induced hypersensitivity reaction or drug-related rash requiring treatment in a hospital, ASM drug-associated rash involving conjunctiva or mucosa, or >1 maculopapular rash requiring discontinuation	4.Taking retigabine (ezogabine) within the past year 5. History of any drug-induced rash or hypersensitivity reaction 6. First-degree relatives with a serious cutaneous, drug-induced adverse reaction
Median % seizure reduction from baseline ^d^		ITT population (primary endpoint) CNB 200 mg (↓55%) * vs. placebo (↓21%)	mITT population (FDA primary endpoint) CNB 400 mg (↓55%) * vs. CNB 200 mg (↓55%) * vs. CNB 100 mg (↓35%) * vs. placebo (↓24%)	NA
Responder rate, % of patients ^e^		• ITT population (secondary endpoint) CNB 200 mg (50%) * vs. placebo (22%) • Post hoc analysis (maintenance phase) CNB 200 mg (62%) * vs. placebo (32%)	mITT-M population (EMA primary endpoint) CNB 400 mg (64%) * vs. CNB 200 mg (56%) * vs. CNB 100 mg (40%) * vs. placebo (25%)	NA
100% seizure reduction during maintenance phase, % of patients		Post hoc analysis CNB 200 mg (28%) * vs. placebo (8%)	Secondary endpoint CNB 400 mg (21%) * vs. CNB 200 mg (11%) * vs. CNB 100 mg (3%) vs. placebo (1%)	NA
Median % seizure reduction by seizure subtype from baseline		ITT population (secondary endpoint) • Focal aware motor seizures CNB 200 mg (↓76%) * vs. placebo (↓27%) • Focal impaired awareness seizures CNB 200 mg (↓55%) * vs. placebo (↓21%) • Focal to bilateral tonic-clonic seizures CNB 200 mg (↓77%) * vs. placebo (↓33%)	mITT-M population (post-hoc analysis) • Focal aware motor seizures CNB 400 mg (69%) * vs. CNB 200 mg (62%) * vs. CNB 100 mg (49%) * vs. placebo (↑11%) • Focal impaired awareness seizures CNB 400 mg (61%) * vs. CNB 200 mg (55%%) * vs. CNB 100 mg (32%) vs. placebo (29%) • Focal to bilateral tonic-clonic seizures CNB 400 mg (83%) * vs. CNB 200 mg (92%) * vs. CNB 100 mg (51%) vs. placebo (33%)	NA
Most common TEAEs, % of CNB patients (occurring in ≥10% of patients with any dose)		• 22% somnolence • 22% dizziness • 12% headache • 11% nausea • 10% fatigue	• 18% (100 mg), 20% (200 mg), 36% (400 mg) somnolence • 17%, 20%, 33% dizziness • 10%, 10%, 10% headache • 12%, 17%, 24% fatigue • 7%, 10%, 15% diplopia	• 28% somnolence • 23% dizziness • 16% fatigue • 11% headache
Serious TEAEs, % of patients		CNB (1.8%) vs. placebo (3.7%)	CNB 400 mg (7.2%), 200 mg (3.6%), 100 mg (9.3%) vs. placebo (5.6%)	8.1%
Hypersensitivity reactions in CNB-treated patients, n of patients		1 (reddening of palms and soles and itching of ears)	3 (1 non-serious pruritic rash with fever, 1 non-serious rash and facial swelling, 1 DRESS)	1
DRESS, n of patients		0	1 (randomized to 200 mg cenobamate with weekly titration)	0
Deaths, n of patients (relationship to study drug)		1 (unrelated, occurred prior to randomization)	0	4 (3 unrelated; 1 remotely related)

Abbreviations: ASM: antiseizure medication; CNB: cenobamate; DB: double-blind; DR: dose-response; DRE: drug-resistant epilepsy; EEG: electroencephalogram; FBM: felbamate; FDA: U.S. Food and Drug Administration; ITT: intention-to-treat; m-ITT: modified intention-to-treat; mITT-M = modified intention-to-treat-maintenance phase; NA: not available/not reported; OLE: open-label extension; PB: phenobarbital; PHT: phenytoin; R: randomized; TEAE: treatment-emergent adverse event; VGB: vigabatrin. ^a^ Treatment-resistant (≥3 seizures per month) despite treatment with one to three ASMs. ^b^ Seizures despite treatment with at least one ASM within the past two years and taking stable doses of one to three concomitant ASMs. ^c^ Initial starting dose of 100 mg/day with a faster titration schedule of 100 mg increments weekly was amended to an initial starting dose of 50 mg/day with a slower up-titration after a blinded review of the first nine patients. ^d^ Based on seizure frequency per 28 days. ^e^ Responder rate defined as ≥50% reduction in seizure frequency. * Significant at 0.05 level.

**Table 6 medicina-59-01389-t006:** Case reports, clinical trials, and observational studies of cenobamate (YKP3089).

Reference	Population	Intervention/Outcome	Comparison	Results/Conclusion	Study Design	Comment
Krauss et al., (2019) [61]	Adult patients with uncontrolled FOS	Safety, efficacy, and tolerability of adjunctive CNB	CNB at dose groups of 100, 200, or 400 mg, or placebo	CNB reduced focal-onset seizure frequency, in a dose-related fashion	MC, DB, R, PC, dose–response	NCT01866111; CO17; N = 437
Trenite et al., (2019) [37]	Adults with photosensitive epilepsy, with/without concomitant ASM therapy	Effect of CNB in patients with PPR to IPS	CNB at dose groups of 100, 250, or 400 mg, or placebo	CNB is a potentially effective product for epilepsy	MC, single-blind	NCT00616148 N = 6
Chung et al., (2020) [60]	Adult patients with uncontrolled FOS	Safety, efficacy, and tolerability of adjunctive CNB	CNB 200 mg or placebo	CNB significantly improved seizure control	MC, DB, PC	NCT01397968; CO13 N = 222
Sperling et al., (2020) [62]	Adult patients with uncontrolled FOS	Safety and tolerability of adjunctive CNB	CNB 12.5 mg/d was initiated and increased at 2-week intervals to 25, 50, 100, 150, and 200 mg/day	CNB was generally well tolerated in the long term, with no new safety issues found	MC, OL	NCT02535091; CO21 N = 1347
Vernillet et al., (2020) [63]	Healthy subjects	Pharmacokinetic characteristics	CNB single (5 to 750 mg) and multiple (50 to 600 mg/day) oral doses or placebo	CNB pharmacokinetic characteristics	R, PC, DB	N = 210
Elizebath et al., (2021) [64]	Adult patients with uncontrolled FOS	Quality of life in epilepsy-31	CNB 100–200 mg/day	Stable treatment responses during CNB treatment. High responders had high scores in quality of life	Two OL extensions of R and PC studies. One OL safety study	Treated at one center for up to eight years. N = 49
French et al., (2021) [65]	Adult patients with uncontrolled FOS	Safety and tolerability of adjunctive CNB	CNB 50–200 mg or placebo	Safety and tolerability of adjunctive CNB treatment	MC, DB, R, PC, multinational	NCT01397968; CO13; N = 149
Rosenfeld, W.E.; Nisman, A.; et al., (2021) [66]	Adult patients with uncontrolled FOS	Efficacy of adjunctive CNB	CNB at dose groups of 100, 200, or 400 mg, or placebo	Reductions in seizure frequency, which was mainly with the 200 and 400 mg/day groups.	DB, PC, PHA	NCT01866111; N = 397
Rosenfeld, W.E.; Abou-Khalil, B.; et al., (2021) [67]	Adult patients with uncontrolled FOS	Efficacy of adjunctive CNB	CNB 12.5 mg/d was initiated and increased at 2-week intervals to 25, 50, 100, 150, and 200 mg/day	Concomitant ASM dose reductions were associated with more patients remaining on CNB	MC, OL, phase 3, PHA	CO21; N = 240
Sander et al., (2021) [68]	Adult patients with FOS	Retention rates	NA	High retention rates	Two R, PC, CNB studies and one OL safety and pharmacokinetic	N = 1844
Sperling et al., (2021) [69]	Adult patients with uncontrolled FOS	Efficacy of adjunctive CNB	CNB 12.5 mg/d was initiated and increased at 2-week intervals to 25, 50, 100, 150, and 200 mg/day	High rates of sustained seizure reduction, with many achieving response early during titration	MC, OL, phase 3, PHA	CO21; N = 240
Yang et al., (2021) [70]	Healthy Japanese subjects	Pharmacokinetics and safety of CNB	CNB at dose groups of 50 mg, 100 mg, 200 mg, or 400 mg	Similar results to the pattern in non-Japanese subjects	R, DB, PC	KCT0002880 N = 32
Abou-Khalil et al., (2022) [71]	Adult patients with uncontrolled FOS	Efficacy of adjunctive CNB	CNB 12.5 mg/d was initiated and increased at 2-week intervals to 25, 50, 100, 150, and 200 mg/day	Efficacy of CNB in patients with DRE despite prior surgery	MC, OL, phase 3, PHA	CO21; N = 240
Aboumatar et al., (2022) [72]	Adult patients with uncontrolled FOS	Efficacy of adjunctive CNB	CNB 12.5 mg/d was initiated and increased at 2-week intervals to 25, 50, 100, 150, and 200 mg/day	A higher percentage of patients with less vs. more frequent seizures at baseline reached zero seizures	MC, OL, phase 3, PHA	CO21; N = 240
Brandt et al., (2022) [73]	Adult patients with uncontrolled FOS	Efficacy of CNB with co-administration of an ASM that is or is not a sodium channel blocker	CNB at dose groups of 100, 200, or 400 mg, or placebo	CNB is effective with or without sodium channel blocker ASMs	MC, DB, R, PC, dose–response	NCT01866111; CO17; N = 437
Connor et al., (2022) [74]	Adult patients with uncontrolled FOS living with a developmental disability	Efficacy and tolerability of CNB	CNB 50–300 mg/day	CNB is effective and was well tolerated	RE medical chart review	N = 28
Darpo et al., (2022) [75]	Healthy adults	Effects of CNB on the QT interval	Therapeutic and supratherapeutic CNB doses	CNB had no relevant effects on electrocardiographic parameters	Single-center, R, DB, PC, parallel-design	N = 108
Elliott et al., (2022) [76]	Adolescents and adults patients with uncontrolled FOS	Real-world application, a history of drug-related rash	CNB 50–300 mg/day	Patients with a history of rash may benefit from CNB	RE medical chart review	N = 45
Klein et al., (2022) [77]	Adult patients with uncontrolled FOS	Long-term efficacy of adjunctive CNB	CNB (target dose, 300 mg/d; min/max, 50/400 mg/d)	Long-term efficacy was sustained during 48 months of CNB treatment. No new safety issues were identified	MC, DB, R, PC	NCT01866111; N = 355
Makridis, K.L.; Bast, T.; et al., (2022) [78]	Pediatric patients with uncontrolled FOS	Efficacy of adjunctive CNB	CNB 50–400 mg/day	CNB is effective and well-tolerated	RE, MC	N = 16
Makridis, K.L.; Friedo, A.; et al., (2022) [79]	Adult patients with Dravet syndrome	Efficacy of adjunctive CNB	CNB 150–250 mg/day	Long-lasting and significant seizure reduction	RE, MC	N = 4
Rosenfeld et al., (2022) [80]	Adult patients with uncontrolled FOS	Efficacy of adjunctive CNB	CNB 12.5 mg/d was initiated and increased at 2-week intervals to 25, 50, 100, 150, and 200 mg/day	Seizure reductions occurred in all focal seizure subtypes with CNB, with the earliest onset in the focal to bilateral tonic-clonic group	MC, OL, phase 3, PHA	CO21; N = 240
Schuetz et al., (2022) [81]	Adult patients with uncontrolled FOS	Adjunctive treatment with CNB is associated with changes in cognitive performance	CNB 50–250 mg/day	Most of the patients showed stable or improved cognitive performance	Prospective observational	N = 59
Steinhoff et al., (2022) [82]	Adult patients with uncontrolled FOS	Efficacy onset and characteristics of time to onset, duration, and severity of the most common treatment-emergent adverse events	CNB 50–200 mg or placebo	Reductions in seizure frequency occurred during titration with initial efficacy observed prior to reaching the target dose	MC, DB, R, PC, PHA, multinational	NCT01397968; CO13; N = 149
Varughese et al., (2022) [83]	Pediatric patients with uncontrolled FOS	Efficacy of adjunctive CNB	CNB 50–400 mg/day	CNB is effective and well-tolerated	RE, MC	N = 21
Agashe et al., (2023) [84]	Pediatric patients with generalized-onset seizures due to generalized or combined generalized and FOS	Efficacy of CNB	CNB 50–200 mg/day	CNB is effective and well-tolerated	RE medical chart review	N = 13
Carlson et al., (2023) [85]	Adult patients with super-refractory status epilepticus	Efficacy of CNB	CNB 200 mg/day	Both patients achieved seizure control	Case report	N = 2
Elakkary et al., (2023) [86]	Adult patients with uncontrolled FOS	Pharmacokinetic interactions between CNB and CLB	CNB 150–200 mg/day	Concomitant administration of CNB and CLB can lead to a substantial increase in serum concentrations of NCLB	RE medical chart review	N = 5 Increased levels of NCLB were associated with positive therapeutic effect, but with increased levels of fatigue
Falcicchio, G.; Lattanzi, S.; et al., (2023) [87]	Adult patients with Lennox–Gastaut syndrome	Efficacy of CNB	CNB 200–300 mg/day	CNB reduced baseline seizure frequency ranged from 25 to 74%, with two patients achieving 50% seizure reduction	RE medical chart review	N = 4
Falcicchio, G.; Riva, A.; et al., (2023) [88]	LAMC3-associated cortical malformations	NA	CNB 300 mg/day	CNB was administered and a partial reduction in seizure frequency	Case report	N = 1
Osborn et al., (2023) [89]	Adult patients with uncontrolled FOS	Pharmacokinetic interactions between CNB and CLB	CNB 25–100 mg/day	Low-dose CLB could be considered in patients with incomplete response to CNB	RE medical chart review	N = 11
Peña-Ceballos et al., (2023) [90]	Adult patients with uncontrolled FOS	CNB’s efficacy and tolerability in a “real-world” severe DRE cohort	CNB 75–350 mg/day	Patients with highly active and ultra-refractory focal epilepsy experienced meaningful seizure outcomes on CNB	RE medical chart review	N = 57 Emergence of adverse events at CNB doses above 250 mg/day
Villanueva et al., (2023) [91]	Adult patients with uncontrolled FOS	CNB’s efficacy and tolerability in a “real-world” Spanish expanded access program	CNB 25–300 mg/day	CNB showed a high response regardless of prior and concomitant ASMs. Adverse effects were frequent but mostly mild-to-moderate, and few led to discontinuation	MC, RE, observational	N = 170
Rosenfeld et al., (2023) [92]	Adult patients with uncontrolled FOS	Mortality and standardized mortality ratio during CNB therapy	CNB 100–400 mg/day	CNB may reduce excess mortality associated with epilepsy	RE medical chart review	N = 2132

Abbreviations: ASM: antiseizure medication; CNB: cenobamate; CLB: clobazam; DB: double-blind; DRE: drug-resistant epilepsy; FOS: focal onset seizure; IPS: intermittent photic stimulation; MC: multicenter; N: number of participants; NCLB: N-desmethylclobazam; OL: open-label; PC: placebo-controlled; PHA: post hoc analysis; PPR: photoparoxysmal-EEG response; R: randomized; RE: retrospective.

**Table 7 medicina-59-01389-t007:** Cenobamate dosing approved according to U.S. Food and Drug Administration protocol ^a.^

Timing		Amount
Initial dosage	Weeks 1 and 2	12.5 mg once daily
Titration regimen	Weeks 3 and 4	25 mg once daily
	Weeks 5 and 6	50 mg once daily
	Weeks 7 and 8	100 mg once daily
	Weeks 9 and 10	150 mg once daily
Maintenance regimen	Week 11 and later	200 mg once daily
Incremental doses after 200 mg/day	Every two weeks	Increase 50 mg once daily until 400 mg once daily
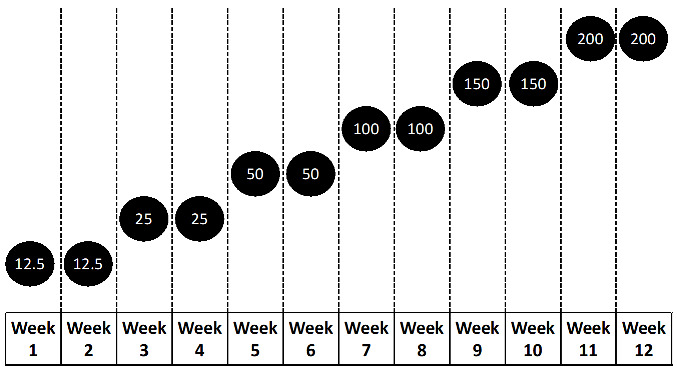		

Observation: Cenobamate maximum dosage recommended is 400 mg once daily. Above cenobamate 250 once daily, there is a significant incidence of side effects. ^a^
https://www.xcopri.com (accessed on 30 May 2023).

**Table 8 medicina-59-01389-t008:** Takeaways by Steinhoff et al. [131] adapted by Rissardo et al.

Who is a candidate for CNB treatment?		Any adults with uncontrolled focal onset seizures.
Specific populations	Elderly	Individuals should be monitored for central nervous system and cognitive adverse events and monitored carefully for potential drug–drug interactions.
	Woman of child-bearing age	Women who are actively seeking to become pregnant or who are pregnant should consider a different ASM.
Who is not a candidate for CNB therapy?		Patients who require an immediate effect from an ASM. Also, individuals with a history of DRESS or severe hypersensitivity reactions to other drugs.
What is the target CNB dose?		CNB 200 mg/day should be the initial target dose. But, initial signs of efficacy have been reported early in the titration period.
When should the pill be taken?		Dose at bedtime to alleviate adverse events.
Counseling patients about common adverse effects		Advise the patients to inform their provider if adverse effects occur. Adverse effects include somnolence (dose-dependent), dizziness (dose-dependent), fatigue (dose-dependent), diplopia (dose-dependent), headache, and nausea.
Adjusting concomitant ASM		This will depend on the dose/concentration of the ASM and the dose of CNB.

Abbreviations: ASM: antiseizure medication; CNB: cenobamate.

## Data Availability

Not applicable.

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
