# Peer review of "Cenobamate (YKP3089) and Drug-Resistant Epilepsy: A Review of the Literature"

_medicina, 2023, doi:10.3390/medicina59081389_

Round 1
Reviewer 1 Report
The present review byRissardo et al. is a helpful and accurate summary that provides a very good overview of the substance cenobamate and can also be a helpful support for clinicians in their daily work.
Clearly structured and well organized in terms of content.
Minor comments:
1) Typo in lin 169 "cenobamatea". Please correct.
2) Please use the internationally recommended term antiseizure medication (ASM) instead of antiepileptic drug (AED)
3) List of drug interactions could extended by more anti seizure medication and oral contraceptives, which is an imprtant topic.
4) Figure 3 is hard to read except the time scale.
Fine.
Author Response
The present review byRissardo et al. is a helpful and accurate summary that provides a very good overview of the substance cenobamate and can also be a helpful support for clinicians in their daily work. Clearly structured and well organized in terms of content. Minor comments: 1) Typo in lin 169 "cenobamatea". Please correct.
Authors: In line 169, there is no mention of cenobamatea. The authors believe that the Reviewer is talking about the title of Table 2, but the a in cenobamatea is in superscript to mention where the information came from.
2) Please use the internationally recommended term antiseizure medication (ASM) instead of antiepileptic drug (AED)
Authors: The authors agree with the Reviewer, and all the “antiepileptic drug (AED)” were modified to “antiseizure medication (ASM).”
3) List of drug interactions could extended by more anti seizure medication and oral contraceptives, which is an imprtant topic.
Authors: The authors agree with the Reviewer's opinion that an increased list would be better, but the authors reviewed the literature thoroughly and included every single drug previously analyzed in the studies from the literature. Therefore, no drugs besides those were already analyzed in the literature.
4) Figure 3 is hard to read except the time scale.
Authors: In Figure 3, the authors summarize the clinical trials throughout time. This figure should be closely analyzed with Table 4. The authors expected the reader to understand that some clinical trials are going on and others that already reported the results. Moreover, the reader should observe interesting studies with potential abuse due to CNB and CNB pharmacokinetics. The pharmacological industry is probably trying to observe if there are any other possible formulations to provide a new administration protocol for this medication or even design a new drug with this outcome.
Reviewer 2 Report
Thank you for inviting me to review this manuscript.
This literature review aimed to elaborate on using Cenobamate (CNB) as an anti-seizure medication (ASM).
The author described thoroughly and widely the use of CNB starting from its history, pharmacokinetics, and mechanism of action and some studies in humans.
However, I have some suggestions:
- The term anti-seizure medication (ASM) is more widely used than anti-epileptic drug (AED) nowadays.
- In the introduction, the author wrote the brand name of CNB. It would be better if the author didn’t write the brand name of the drug.
- Figure 1: “Alkyl-carmabates” was mistyped. It should be “Alkyl-carbamates”
- Line 124-136 explained the discovery of the new method to determine the plasma level of CNB. This paragraph didn’t fit the other paragraphs of this section. The main idea of this section was to describe the history of CNB being the ASM for drug-resistant epilepsy (DRE).
- Table 3: abbreviation CLB for clobazam is more common than CLO.
- Figure 2: what is the meaning of “red line”?
- Table 4: author included 3 studies on healthy people and 1 study on “NA” subjects to know the pharmacokinetic (PK) aspect and potential abuse of CNB. I think these studies did not fit the aim of this literature review. The PK has been explained in 3rd section.
- Table 6: what is the abbreviation for NCLB?
- Figure 4 contains an important message for clinicians. It would be better to change this figure to a Table or BOX that can be easily read.
This Literature review is a complete review of Cenobamate. However, there are so many messages that authors want to deliver. It could be shortened to the most important messages to maintain the engagement of the readers.
Author Response
Thank you for inviting me to review this manuscript. This literature review aimed to elaborate on using Cenobamate (CNB) as an anti-seizure medication (ASM). The author described thoroughly and widely the use of CNB starting from its history, pharmacokinetics, and mechanism of action and some studies in humans. However, I have some suggestions:
- The term anti-seizure medication (ASM) is more widely used than anti-epileptic drug (AED) nowadays.
Authors: The authors agree with the Reviewer, and all the “antiepileptic drug (AED)” were modified to “antiseizure medication (ASM).”
In the introduction, the author wrote the brand name of CNB. It would be better if the author didn’t write the brand name of the drug.
Authors: The authors removed the brand names Xcopri® and Ontozry®
Figure 1: “Alkyl-carmabates” was mistyped. It should be “Alkyl-carbamates”
Authors: The authors corrected the name from “Alkyl-carmabates” to “Alkyl-carbamates.”
Line 124-136 explained the discovery of the new method to determine the plasma level of CNB. This paragraph didn’t fit the other paragraphs of this section. The main idea of this section was to describe the history of CNB being the ASM for drug-resistant epilepsy (DRE).
Authors: We would like to maintain this paragraph. The main idea of the chapter is to present historical aspects. The authors found it interesting that new methods were used to evaluate serum CNB. And they are described in the manuscript in the sequential order of development.
Table 3: abbreviation CLB for clobazam is more common than CLO.
Authors: We agree with the reviewer and modified this according to his/her request. We first wrote CLO because it was already described in the literature.
Figure 2: what is the meaning of “red line”?
Authors: The area regarding inside the redline there is no meaning in pharmacology. It is only its peaks that should be assessed. The SwissADME tool was described accordingly by the authors that developed this method.
Table 4: author included 3 studies on healthy people and 1 study on “NA” subjects to know the pharmacokinetic (PK) aspect and potential abuse of CNB. I think these studies did not fit the aim of this literature review. The PK has been explained in 3rd section.
Authors: The main idea of Table 4 is to provide all the clinical trials associated with cenobamate. The numbers are accordingly they are described in the ClinicalTrials.gov database. Furthermore, the inclusion of the clinical trials in the table is for the reader to observe throughout time what were the indications of cenobamate and the number of participants enrolled in the studies. Also, this can provide the information of possible trends of investigation with this medication.
Table 6: what is the abbreviation for NCLB?
Authors: We would like to thank the reviewer for this close observation. We included in the abbreviations “NCLB, N-desmethylclobazam”
Figure 4 contains an important message for clinicians. It would be better to change this figure to a Table or BOX that can be easily read.
Authors: The main idea of this figure was to summarize the interactions of CNB and other ASMs. But, for a better understanding, the reader should assess Table 3, which has a more precise description.
This Literature review is a complete review of Cenobamate. However, there are so many messages that authors want to deliver. It could be shortened to the most important messages to maintain the engagement of the readers.
Authors: We would like specially thank the reviewer for his/her precise observations that we believe improved the quality of the manuscript. The authors primarily wanted to write a thorough revision of the literature regarding cenobamate, they primarily extracted all the manuscripts in Pubmed related to this drug. Therefore, the authors believe this is the most extensive and complete review regarding cenobamate; all the articles were read, and important messages were extracted.